# Enhancing Speech Emotions Recognition Using Multivariate Functional Data Analysis

**Matthieu Saumard** 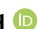

LabISEN Yncréa Ouest, VISION-AD Team, 20 rue Cuirassé Bretagne, 29200 Brest, France;
matthieu.saumard@isen-ouest.yncrea.fr

**Abstract:** Speech Emotions Recognition (SER) has gained significant attention in the fields of human–computer interaction and speech processing. In this article, we present a novel approach to improve SER performance by interpreting the Mel Frequency Cepstral Coefficients (MFCC) as a multivariate functional data object, which accelerates learning while maintaining high accuracy. To treat MFCCs as functional data, we preprocess them as images and apply resizing techniques. By representing MFCCs as functional data, we leverage the temporal dynamics of speech, capturing essential emotional cues more effectively. Consequently, this enhancement significantly contributes to the learning process of SER methods without compromising performance. Subsequently, we employ a supervised learning model, specifically a functional Support Vector Machine (SVM), directly on the MFCC represented as functional data. This enables the utilization of the full functional information, allowing for more accurate emotion recognition. The proposed approach is rigorously evaluated on two distinct databases, EMO-DB and IEMOCAP, serving as benchmarks for SER evaluation. Our method demonstrates competitive results in terms of accuracy, showcasing its effectiveness in emotion recognition. Furthermore, our approach significantly reduces the learning time, making it computationally efficient and practical for real-world applications. In conclusion, our novel approach of treating MFCCs as multivariate functional data objects exhibits superior performance in SER tasks, delivering both improved accuracy and substantial time savings during the learning process. This advancement holds great potential for enhancing human–computer interaction and enabling more sophisticated emotion-aware applications.

**Keywords:** speech emotion recognition; functional data; MFCC

## 1. Introduction

Effective emotion recognition is crucial for facilitating natural and meaningful interactions between humans and robots. Humans express their emotions through various channels, including facial expressions, body language, and speech. Among these modalities, speech plays a pivotal role in emotion recognition due to its ability to convey rich emotional information.

Speech emotion recognition is a challenging task, primarily due to the inherent variability in emotional expressions. Numerous classifiers have been employed for this purpose, with SVMs being the most commonly used. SVMs are supervised learning algorithms capable of learning decision boundaries between different classes of data. They excel in handling high-dimensional data such as speech and can learn complex decision boundaries.

SER is a vital field of research that has gained significant attention due to its potential applications in various domains, including human–computer interaction, affective computing, virtual agents, and intelligent systems. The ability to accurately recognize and interpret emotions conveyed through speech opens up numerous opportunities for enhancing user experiences and improving the efficiency of human–technology interactions. One important aspect of SER that researchers have been focusing on is reducing the learning time required for effective emotion recognition.

One notable application of SER with low learning time processes is in improving human–robot interaction. Robots and virtual agents equipped with the ability to perceive and respond to human emotions can establish more natural and empathetic connections with users. By reducing the learning time required for emotion recognition, robots can quickly adapt to users' emotional states, allowing for more personalized and engaging interactions. This has significant implications in fields such as healthcare, education, customer service, and entertainment, where empathetic and responsive robots can enhance the overall user experience.

Another important application of SER with low learning time processes is real-time emotion monitoring. Traditional emotion recognition systems often require time-consuming training and calibration processes, limiting their ability to provide instantaneous feedback on emotional states. With low learning time processes, SER systems can analyze speech signals in real-time, enabling applications such as emotion-aware virtual assistants, emotion-based feedback systems, and emotion monitoring in high-stress environments (e.g., call centers, emergency response settings). Real-time emotion monitoring can assist in improving communication, emotional well-being, and decision-making processes.

SER with low learning time processes holds promise in assisting individuals with emotional disorders. By accurately recognizing and monitoring emotional states, these systems can provide valuable insights and support for individuals suffering from conditions such as depression, anxiety, and autism spectrum disorders. Low learning time processes enable the development of portable and accessible emotion recognition tools that can be integrated into smartphones, wearable devices, and other assistive technologies. Such applications can empower individuals by providing real-time emotional feedback, personalized interventions, and timely support.

The entertainment industry can also benefit from SER with low learning time processes. By analyzing speech emotions in real-time, content creators can develop immersive and interactive experiences that adapt to users' emotional responses. Emotion-aware video games, virtual reality experiences, and interactive storytelling platforms can dynamically adjust their narratives, characters, and gameplay elements based on users' emotional states. This creates a more engaging and personalized entertainment experience, allowing users to become active participants in the content creation process.

In the subsequent sections, we provide a detailed description of our proposed method. We evaluate its performance on two databases, EMO-DB and IEMOCAP, and present comparative results with other approaches. Finally, we conclude and discuss the outcomes in the closing section.

## 2. Literature Review

In recent years, speech emotion recognition has witnessed significant progress with the advent of deep learning techniques [1–3], including Convolutional Neural Networks (CNNs), transformers, attention mechanisms, and self-supervised learning methods [4–8].

Parallel to these advancements, Functional Data Analysis (FDA) has emerged as a vibrant research field within the statistical community, following pioneering monographs by [9,10]. FDA finds extensive applications in various scientific domains, such as astronomy, chemometrics, health, and finance [11–13]. While the core objects of FDA are curves or functions within a separable Hilbert space, it also encompasses the statistical study of random variables in more general function spaces like Banach space [14] and curves on manifolds [15].

The fundamental frequency curve of an utterance has been previously considered as a functional object [16,17]. Notably, Ref. [18] analyzed dialect sound variations across Great Britain using a spatial modeling approach with MFCCs. Building upon these foundations, this article extends the application of functional data analysis to SER.

In our approach, we interpret each coefficient of MFCC as a functional data variable, allowing us to establish a correspondence between a speech recording and a multivariate

functional data object. The number of covariates in the multivariate functional data object corresponds to the number of coefficients used from the MFCC.

To facilitate the comparison of functional data objects with varying durations across samples, we preprocess the MFCCs by applying a resizing method. Additionally, by considering pitch and a predefined number of MFCCs of the same length across all samples, we leverage a functional machine learning algorithm, specifically a functional SVM, with each MFCC treated as a functional object.

## 3. Method

### 3.1. Extraction of Features

Let us recall the basic principle to calculate the MFCC with Figure 1. With at hand a raw audio signal data representing by a time series $s(t)$ for $t = 1, \ldots, T$. We can consider that $s$ is defined for $t \in \mathbb{Z}$ by adding 0 to a non-value. Let $w_M(t)$ for $t \in \mathbb{Z}$ be a window function of width $M$, we can define the spectrogram of the audio signal $s(t)$ by

$$\text{Spec}(t, \omega) = |\sum_{u=1}^{T} s(t-u) w_M(u) \exp(-i\omega u)|, \tag{1}$$

for $t = 1, \ldots, T, \omega \in [0, 2\pi]$.

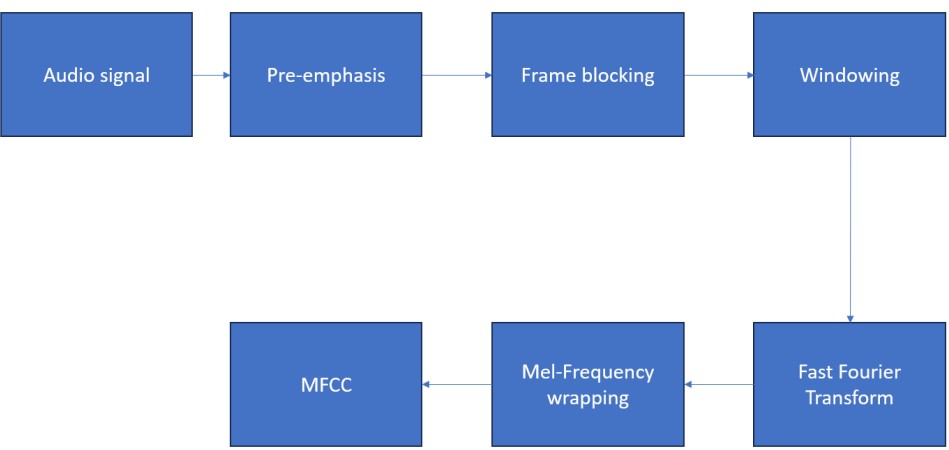

**Figure 1.** Calculation of MFCC.

Hence, we can define the Mel spectrogram, which is a filtered version of the spectrogram to represent human ear auditory system:

$$\text{MelSpec}(t, f) = \sum_{k=0}^{N-1} \text{Spec}\left(t, \frac{2k\pi}{N}\right) b_{f,k} \tag{2}$$

for $f = 0, \ldots, F$, with $b_{f,k}$ representing the set of the Mel-scale filter bank. Recall that the Mel scale is $m = 2595 \log_{10}(1 + \frac{f}{700})$. The MFCCs are then

$$\text{MFCC}(t, m) = \frac{1}{F} \sum_{f=0}^{F} \log(\text{MelSpec(t,f)}) \exp(i(2\pi \frac{m-1}{F+1})f), \tag{3}$$

for $m = 1, \ldots, n_{MFCC}$.

Note that there exist variations of the definition of MFCC in the literature. For example, we can use a Discrete Cosine Transform (DCT) to return to the time scale.

Examples of MFCCs can be viewed in Figure 2 corresponding in case I and II to initial MFCCs (top of both cases) with different duration $T$. In order to compare MFCCs, we choose to resize the MFCCs to a desired size by applying the resize function from cv2 package (OpenCV) in Python. In case I of Figure 2a, we have a greater target time than the

effective duration of the MFCCs. In case II of Figure 2b, the target time is lower than the effective duration of the MFCCs. We choose to not modify the number of MFCCs. In both cases, we plot the intial MFCC and the resized MFCC.

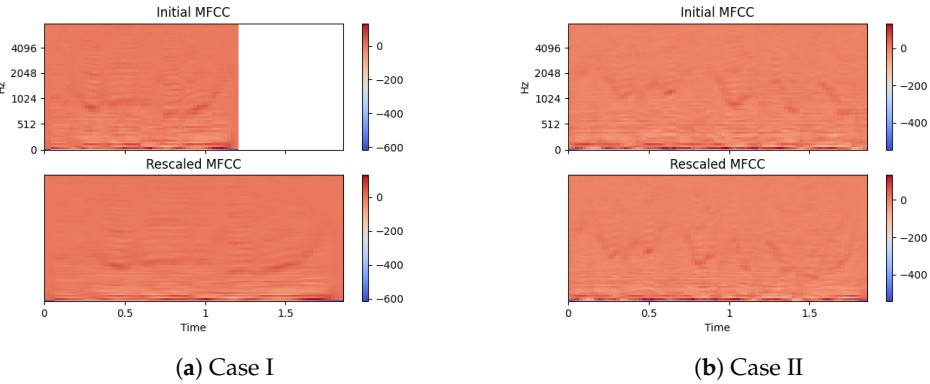

(**a**) Case I

(**b**) Case II

**Figure 2.** Resizing MFCC to size $80 \times 80$.

### *3.2. Functional Features*

Let $\{M_l | l = 1, \dots, n\}$ be the resized MFCC of shape $(n_{MFCC}, m)$ where $n$ is the sample size of the dataset, $n_{MFCC}$ is the number of coefficients of the Discrete Cosine Transform of the spectral envelope (the number of MFCCs) and $m$ is the number of discretization points of time, which is constant over the dataset thanks to the resizing method.

Define $M_l^i$, for $l \in \{1, \dots, n\}$ and $i \in \{1, n_{MFCC}\}$, the evolution of the $i$-th coefficient of the $l$-th data across time. In practice, the evolution is sufficiently smooth to consider $M_l^i$ as a smooth function. Then, for each raw audio signal of index $l$, we have a multivariate functional data $M_l^i(t)$ observed at $m$ regular discretization points $(t_1, \dots, t_m)$, where the number of covariates is $n_{MFCC}$.

Then, during the registration step of the functional data, we operate a soft smoothing of the function in order to convert raw discrete data points into a smoothly varying function. We have constructed $M_l(t)$, a multivariate functional data object for each raw audio signal of index $l$. The next step is to learn these functional features through an intensely used supervised machine learning method, namely SVM, but in a modified version called functional SVM, which is the subject of the next section.

### *3.3. Classification*

Figure 3 shows a schematic of the proposed method with its different parts. First, we compute the MFCCs. Secondly, we resize them to a desired shape. Finally, we can use a SVM to classify the associated emotion.

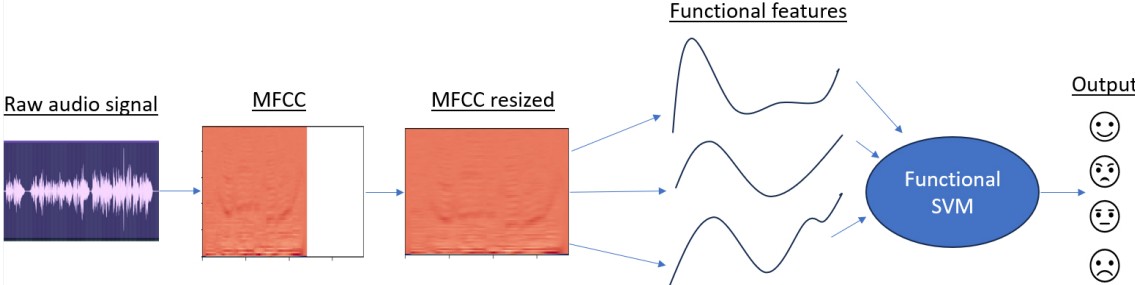

**Figure 3.** Proposed method.

We refer the reader to the paper of [19], which presents an extension to functional data features of the classical SVM algorithm.

We choose a radial basis for the kernel $K$ of the SVM. $K(u, v) = exp(-\gamma |u - v|^2)$. The tuned parameters of the SVM are the cost (the C constant of the regularization term in

the Lagrange formulation) and the $\gamma$ parameter used in the radial basis. We optimize the method on a grid with a cost of 5 and $\gamma = 0.001$. We use these parameters in the two databases of the following section.

In order to incorporate functional features as inputs into the SVM, it is necessary to adapt the SVM algorithm. The approach chosen in this study involves decomposing each functional covariate into a basis representation. Subsequently, the coefficients obtained from the decomposition are utilized as inputs for a conventional SVM classifier.

To conclude this section, we enumerate the different steps to perform a speech emotions recognition task with our approach.

1. Data: Raw audio signal,
2. Compute the MFCC,
3. Resize each MFCC,
4. Use each MFCC as functional features,
5. Learn the functional SVM,
6. Deploy the model for output.

## 4. Application

### 4.1. Databases

EMO-DB (Berlin Database of Emotional Speech) [20] is a publicly available German speech database containing speech recordings of seven emotions: sadness, anger, happiness, fear, disgust, boredom and neutrality. The entire dataset contains 535 speeches in german.The database is created by the Institute of Communication Science, Technical University, Berlin, Germany. Ten professional speakers (five males and five females) participated in data recording.

The dataset IEMOCAP (Interactive Emotional Dyadic Motion Capture) [21] contains approximately 12 h of speech from 10 speakers. The literature selects a total of 5531 utterances that are labeled with one of five categories: Happy, Angry, Neutral, Sad, and Excited. This set is reduced to four categories by merging Excited and Happy into a single category.

Both IEMOCAP and EMO-DB have significantly contributed to advancing the field of speech emotion recognition. They have been widely used as benchmark datasets for developing and evaluating various emotion recognition algorithms and models. The availability of these databases has facilitated comparative studies, the exploration of new methodologies, and the development of robust and accurate emotion recognition systems.

It is imperative to emphasize that solely emotions were employed during both the learning and testing phases. Consequently, metadata elements such as language, gender, actor identity, among others, were not incorporated into the analysis.

### 4.2. Results

We calculate two metrics, namely WA for weighted accuracy (overall accuracy) and UA for unweighted accuracy (average of the recall). We perform 10-fold cross validation on each dataset. So, the presented results are the mean WA and mean UA along with their standard deviations in parentheses. The results in Table 1 are for the dataset EMO-DB with four emotions (happiness, anger, neutral and sadness). Table 2 presents the results with all the seven emotions. The fundamental frequency f0 curve can also be considered as a functional data object, so we decide to compare the results with or without this feature adding to MFCC features. The lines corresponding to MFCC+ Pitch represent the method that adds the fundamental frequency curve. Different sizes have been considered, we present the two relevant sizes, namely 60, which indicates that the array of MFCCs has been resized to $60 \times 60$ and 80 to $80 \times 80$.

We have chosen a cubic B-spline basis (commonly used in FDA, see [9]) to decompose the functional features.

**Table 1.** Results for EMO-DB with 4 emotions.

| Methods | | WA | UA |
|---------|----|----|----|
| MFCC + Pitch | 60 | 0.909 (0.0670) | 0.914 (0.0587) |
|  | 80 | 0.906 (0.0661) | 0.908 (0.0593) |
| MFCC | 60 | 0.906 (0.0646) | 0.913 (0.0572) |
|  | 80 | 0.912 (0.0561) | 0.911 (0.0506) |

**Table 2.** Results for EMO-DB with 7 emotions.

| Methods | | WA | UA |
|---------|----|----|----|
| MFCC + Pitch | 60 | 0.858 (0.0489) | 0.864 (0.0419) |
|  | 80 | 0.834 (0.0568) | 0.836 (0.0574) |
| MFCC | 60 | 0.858 (0.0528) | 0.864 (0.0405) |
|  | 80 | 0.832 (0.0457) | 0.835 (0.0448) |

Let us comment on these three tables of results. The best reshaping size is 60 for all the seven emotions in EMO-DB with a gain of 2% comparing with the size 80. In the case of the four emotions, there is no important gain in favor of one method. The best UA is for 60 and the best WA is for 80, but the results are similar. Concerning the result on IEMOCAP database (Table 3), a pre-result study has been made to select the size and it is $80 \times 80$, which has the best result. The fact that the mean duration of speeches in IEMOCAP database is higher than the one in EMO-DB database is an argument in favor of a higher reshaping size for the IEMOCAP database than for EMO-DB database.

**Table 3.** Results for IEMOCAP with 4 emotions and size 80.

| Methods | WA | UA |
|---------|----|----|
| MFCC + Pitch | 0.648 (0.0167) | 0.654 (0.0193) |
| MFCC | 0.652 (0.0134) | 0.658 (0.0171) |

### 4.3. Effects of the Resizing Method

In order to compare multivariate functional objects, the ranges of each functional variables of the object must correspond. It is then natural to resize all the MFCCs to the same size. The chosen number of MFCCs is an important parameter of our procedure. An insufficient number of MFCC leads to a loss of information and accuracy. And the other hand, too many MFCCs do not add relevant information and increase the computation time. The other parameter we must take in account is the duration of the audio sample. The mean duration time in the IEMOCAP database is higher than in the EMO-DB database. An idea with a very high duration audio sample would be to cut in chunks of the audio file and to process them with our approach. Comparing the size on the results with the EMO-DB database, it is not clear which number of MFCCs must be chosen between 60 and 80. The results are very similar. But in the IEMOCAP database, the results fall dramatically when we change to 60 MFCCs. This is caused by two factors: the mean duration time and the number of samples (which are higher in the IEMOCAP database).

### 4.4. Comparison with Other Methods

It is important to note that previous functional methods, which solely consider Pitch as a functional feature, do not achieve the same performance results as our approach. In order to assess the effectiveness of our method, we compared our results with state-of-the-art (SOTA) deep learning methods, namely [22] for the IEMOCAP database and [4] for the EMO-DB database, which were the top-performing methods in 2021 for these two databases.

In the case of the EMO-DB database, the article by [4] achieved an overall accuracy of 97.7%, whereas our approach achieved 85.8%. This indicates that our method, although slightly lower in accuracy, still demonstrates competitive performance compared to the SOTA approach of [4].

Similarly, when comparing our results with the findings of [22] on the IEMOCAP database, their weighted accuracy (WA) and unweighted accuracy (UA) were both reported as 83.3%, whereas our WA and UA were 65.2% and 65.8%, respectively. Although our results are lower than those of [22], it is important to consider the training time. Our method achieves remarkable efficiency, with a mean training time of only 11 s on a standard laptop, while [4] reported a training time of approximately one week in their article.

In summary, while our method may have slightly lower accuracy compared to the SOTA approaches of [4,22], its notable advantage lies in its significantly reduced training time. This makes our approach highly efficient and practical, especially when considering real-world applications where quick adaptation to new data is required.

## 5. Conclusions and Discussion

In conclusion, our approach of treating Mel-Frequency Cepstral Coefficients (MFCC) as functional data objects offers a promising solution for speech emotion recognition tasks. The methodology we propose not only demonstrates high efficiency, but also proves to be well-suited for implementation on small devices, such as Field-Programmable Gate Arrays (FPGAs).

Compared to deep learning methods, our approach significantly reduces the learning time required for emotion recognition, making it particularly suitable for scenarios where new data becomes available at different intervals. This capability is valuable in applications where real-time or near real-time emotion recognition is essential, such as interactive systems or online platforms. Additionally, the reduced learning time also contributes to resource efficiency, making our method more practical and cost-effective for deployment on devices with limited computational capabilities.

Another advantage of our approach is its compatibility with small devices. By utilizing MFCC as functional data objects, our method can be implemented on resource-constrained platforms, enabling on-device emotion recognition without relying heavily on external computational resources. This expands the possibilities for deploying emotion recognition systems in various settings, including wearable devices, smartphones, and embedded systems, thereby enhancing human–computer interaction in a wide range of contexts.

While our approach exhibits notable performance, there is room for further improvement. One potential avenue for enhancing accuracy is the utilization of alignment methods for each coefficient of MFCC. Aligning the functional representations of MFCC could potentially address variations in speech duration and improve the alignment of emotional features across different utterances. This refinement has the potential to further boost the accuracy and reliability of our approach, contributing to more precise emotion recognition.

In summary, our approach offers a compelling alternative for speech emotion recognition, offering fast learning times, adaptability to small devices, and potential accuracy improvements through alignment techniques. With further research and refinement, our methodology holds great promise for advancing emotion-aware technologies and facilitating more natural and empathetic human–computer interactions.

**Funding:** This research received no external funding.

**Data Availability Statement:** The EmoDB database is freely avalaible at http://emodb.bilderbar. info/docu/#emodb (accessed on 20 July 2023). The IEMOCAP database is freely avalaible upon request at https://sail.usc.edu/iemocap/ (accessed on 20 July 2023).

**Conflicts of Interest:** The author declares there is no conflict of interest.

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
