# Peer review of "Enhancing Speech Emotions Recognition Using Multivariate Functional Data Analysis"

_2504-2289, doi:10.3390/bdcc7030146_

Round 1

Reviewer 1 Report

The manuscript is quite interesting to read and the results are worth publication. I have only a few technical notes and one general note.

The technical notes are simple:

1) "Discrete Cosinus transform" should be "Discrete Cosine transform"

2) Please, avoid placing a hyphen within $...$ in the LateX code in the following cases: $i-$th, $l-$th. Use $i$-th, $l$-th instead. The text would look way neater then.

The main point to address is as follows:

You train your models based on two different datasets. Would it be possible to add some comparison of the results obtained using model 1 on dataset 2 and vice versa? For instance, by limiting the output to four emotions only or something of that kind? If yes, please, add this -- your models are quite fast to obtain the result. If no, please, comment on why this is not possible.

See the note above on Cosinus -> Cosine.

Reviewer 2 Report

  1. Overall, I am satisfied and convinced with the presented research.  However, in interest of improving the quality of the research work, it is recommended that the author considers the following comments meant to further fine tune and improve the research work and its presentation.
  2. The corresponding long forms should accompany all the ‘first usages’ of abbreviations (in abstract and the remaining manuscript).  At the same location, the words of the long form should be suitably written in Title Case.  Either the style of ‘long form followed by the abbreviation’ (preferably) or the ‘abbreviation followed by the long form’ should be consistently used throughout the manuscript.  After the abbreviation has been defined at the first instance, the subsequent text of the manuscript should not unnecessarily mention the abbreviation and long form again, and rather only the abbreviation should be used.  Particularly, please note that the ‘first usage’ concept applies to each of the two sections namely abstract and the remainder of the text.
  3. It is highly recommended that the authors include comments on the metadata of the datasets used.  This is important because the claimed results may not hold true in general, and may be suited for the particular cases considered for experimentation through the said databases.  In particular, the authors are recommended to mention about the language or languages (including dialects, if any) of the datasets, the geographical region(s) of the speakers, etc.  This is also important as the expression of emotion differs from language to language, and region to region.  Many a times mere emphasis on a word conveys the sentiment!  It will be highly interesting to know the author’s comments on this.
  4. Generally, the table captions are above tables while the captions of figures are below them.
  5. The number (and types too) of emotional states in both datasets are different.  Author has presented four states as output in Fig. 3.  How has the author reconciled all these differences?
  6. Author has claimed the reduction in time using the proposed approach.  It is notable that such comparisons are relative.  Inspite of the claim, the author has neither included any such comparison table, nor any quantified values of time anywhere in the manuscript.

Overall, the usage of English language is acceptable.

Round 2

Reviewer 2 Report

The suggestions of previous review round have been well-considered by the authors.

The manuscript can be accepted in present form.